# Shedding Light on the Origin of Egyptian Sheep Breeds by Evolutionary Comparison of Mitochondrial D-Loop

**DOI:** 10.3390/ani12202738

**Published:** 2022-10-12

**Authors:** Agnès Germot, Muhammad Gamal Khodary, Othman El-Mahdy Othman, Daniel Petit

**Affiliations:** 1LABCiS, University of Limoges, UR 22722, F-87000 Limoges, France; 2Integrative Biosciences (IBS) Department, Tuskegee University, 1200 W Montgomery Rd., Tuskegee, AL 36088, USA; 3Cell Biology Department, National Research Center, El Buhouth St. (El Tahrir St.), Dokki, Giza 12311, Egypt

**Keywords:** fat-tail introgression, mitochondrial control region, Egyptian breed, phylogenetic analysis

## Abstract

**Simple Summary:**

Egypt is a carrefour between North African Maghreb countries, East Tropical Africa, Arabian Peninsula, and Near East countries including Turkey. The aim of the present work was to describe the genetic relationships between several domestic sheep populations of these regions. All of the present-day Egyptian breeds are coarse woolen and fat-tailed, but archeologists indicate that this tail form was acquired in the Near East later from populations with a thin tail. To test this idea, we used a phylogenetic-derived program to compare the control region of mtDNA of 37 breeds with fat or thin tails. We showed that most breeds seemed to fit with the archeologist hypothesis, whereas one breed indicates a direct migration of a fat-tailed breed from Turkey. Unexpectedly, one of the breeds from South Egypt was strongly linked to the thin-tailed desert breeds of Sudan, raising the question of the events leading to this situation.

**Abstract:**

(1) Background: It has been recognized that the origin of fat-tailed sheep occurred within coarse wool breeds and that this character was introgressed several times into thin-tailed populations. However, no study has investigated this idea for Egyptian breeds using mtDNA analyses. (2) Methods: Using new sequences of the control region, we constructed a database of 467 sequences representing 37 breeds including fat- and thin-tailed ones with 80 Egyptian individuals belonging to six local breeds (Barki, Fallahi, Ossimi, Rahmani, Saidi, Sohagi). The phylogenetic tree obtained with the maximum likelihood method was submitted to the Newick Extra program to count the direct and indirect links between the individuals of each breed. (3) Results: Several Egyptian breeds were strongly connected to “primitive” thin-tailed breeds from Europe, indicating a clear genetic background of the “thin tail” breed type that supports the view of archeologists. In several cases, we suspected Western Asian breeds to be involved in the introgression of the fat tail character. In contrast, the Ossimi breed showed a high affinity to a fat-tailed breed of Western Asia, suggesting a direct migration and no thin tail ancestors. The Saidi is unique as our analyses revealed its strong connection with thin-tailed Sudanese breeds.

## 1. Introduction

The domestication of sheep from Mouflon ancestors has been intensively studied by archeologists and more recently by geneticists using molecular tools. One of the first bone remains was found in the Taurus Mountain in Southeast Anatolia (Turkey), as reported by Peters et al. (2005) [1] and dated around 10,500 years BP. The analysis of mitochondrial DNA identified the Asian Mouflon (*Ovis gmelini subspecies anatolica*, sometimes named *Ovis orientalis anatolica*) as the closest modern taxon to the domestic sheep (*Ovis aries*) [2,3]. In contrast, the Urial (*O. vignei*) were excluded from the potential ancestors to the domestic species [4]. The European Mouflon (*O. aries musimon*) is now interpreted as a feral population relict of the first expansion of *Ovis aries* such as several domestic populations living in the North of Europe (e.g., Soay, Orkney, Finnsheep, Gotland-Gute, among others) [5]. According to the same authors from the insertion polymorphism of retroviruses, most breeds nowadays are linked to a second domestication process, based on the selection of interesting characters including wool production. As a result, domestic sheep have spread in all inhabited continents due to their ability to cope with plants of low nutritional value and live in harsh environments.

Breeds nowadays are usually classified according to their fleece, which can be made of hair or wool, itself being coarse, semi-fine, or fine [6]. Within the coarse-woolled group, the tail is another character, allowing for the distinction between thin-, fat-tailed, or fat-rumped breeds [7]. Fat-tailed breeds constitute around 25% of the sheep population worldwide. Their dominance in the Fertile Crescent supports the views of several authors [8,9,10] who have argued that the adaptive interest of this fat deposition is to resist the extreme contrasting seasons, limited quality and quantity of food, and long migration. The genetic basis of the genes selected to these adaptations has been investigated [7,10,11]. The spread of domestic sheep in North Africa could have begun with thin-tailed population from the Fertile Crescent via the Isthmus of Suez around 7000 years BP [12,13]. Ryder [14] reported that the first record of fat-tailed animals was attested from a stone vessel picture of 5000 years BP discovered in Uruk (Iraq). Fat-tailed populations came into North and Northeast Africa in a second wave via the Isthmus of Suez and the Horn of Africa across the strait of Bab-el-Mandeb, respectively. However, the changes were not always in the direction of thin tail to fat tail. As demonstrated through an investigation carried out using SNP markers, Ahbara et al. [11] indicated that the different tail forms in Ethiopian breeds may result from the introgression of the thin phenotype in ancestral fat-tailed breeds, or from the direct migration of thin-tailed populations.

The population size of sheep in Egypt was 5.69 million head in 2019, exceeding that of cattle, goats, or buffaloes [15]. In Egypt, a dozen breeds are settled in the territory and all are fat-tailed because of the preference of livestock keepers for breeds of this tail phenotype, better adapted to desert-like conditions [15,16,17]. Moreover, there is a tradition in Arab cooking to consume the tail stored fat for its unique quality [18]. Their fleece is a coarse wool [19,20,21]. From analyses using mtDNA sequencing [22,23,24] and microsatellites [25], the relationships between the major breeds of Egypt were investigated but their connections with other breeds were limited to Algerian and some fat-tailed European breeds.

Using the analysis of the control region of mitochondrial DNA, the aim of this work was to test whether the fat tail of Egyptian sheep breeds was acquired through an introgression affecting thin-tailed populations (Hypothesis 1, according to the view of archeologists). In this case, two questions are raised. First, are there breeds today that would be genetically close to the ancestral thin-tailed population? Second, which populations were involved to introduce the genes driving the fat tail phenotype? In contrast, are there Egyptian breeds resulting from the direct migrations of fat-tailed populations issued from Western Asia (Hypothesis 2)? Otherwise, given the progressive spread of sheep populations from a center situated in the Fertile Crescent, the question of the involvement of Egyptian breeds as ancestors of other breeds situated southward and westward of Egypt are raised.

## 2. Materials and Methods

### 2.1. Animals

Among the dozen breeds of Egypt, the six following ones were studied. Their synthetic description was taken from [15,17,19,20,21] with illustrations in [16,17], completed by the Appendix A for Fallahi breed and the addition of Barki ram.

The Barki is located in the desert area on the coastal zone in the northwest of Egypt between Alexandria and the northeastern part of Libya, where the Barqa province gives its name. It is similar to the fat-tailed sheep of Libya. The fleece on the body is white but has a brown or black head, and the coarse wool is superior in weight and quality to that of the Rahmani and Ossimi, however, it has a smaller size relative to these breeds.

The Rahmani is the largest among all of the Egyptian breeds, with a long straight brown wool. It is settled in the north and middle of the Nile Delta, and named after Rahmania in the Beheira governorate. It is said to have been introduced in Egypt in the 19th century.

The Ossimi has an intermediate size between the two previous breeds. Its fleece is white. Its cradle is the southern part of the Nile Delta and its name comes from Ossim, a village near Cairo. Its population is spread in Middle and Upper Egypt due to a wider range of adaptability than Barki and Rahmani.

The Saidi has a long fat tail and a generally dark brown fleece. Its breeding area is in the Upper Egypt South of Assiut and some authors consider it as the oldest of the Egyptian breeds.

The Sohagi has a size close to Barki. It lives in South Egypt and is named after the Sohag governorate. It is of crucial importance for the livelihoods of farmers living in the poorest areas of the country.

The Fallahi inhabits parts of the Nile Delta outside the territories of the Rahmani and Ossimi breeds, and is found particularly in the northern parts of the Sharkiya and Garbiya provinces. They are usually brown and their fat tail tapers from a broad base.

### 2.2. Sequencing

We used the genomic DNA extracts sampled by Othman et al. in 2014 [22] and 2018 [24] from the blood of animals of both sexes to achieve new sequencing. For this purpose, PCR was performed using the combination of the forward primers, CR0F (5′-TGGTCTTGTAAACCAGAGAAGGAG-3′) designed using Primer-Blast from NCBI tools, or CR2F (5′-GAAGTTCTACTTAAACTATTCCCTG-3′) and the reverse one CR3R (5′-GATGCTCAAGATGCAGTTAAGTCC-3′), both primarily designed in [26]. The first couple CR0F/CR3R allows for the amplification of a portion of the mtDNA control region from 15,326 bp to 16,390 bp of the *Ovis aries* Merinolandschaf mitogenome (NC_001941 in Genbank) and in the CR2F/CR3R one, the amplification was from 15,412 bp to 16,390 bp. PCR reactions were carried out in a final volume of 20 μL PCR mix containing 100 ng of genomic DNA, 4 μL of 5X buffer, 1.6 µL of 2.5 mM dNTPs, 10 pmol of each primer, and 0.4 U of Phusion High-Fidelity DNA polymerase (ThermoFisher Scientific, Waltham, MA, USA). After 30 s of denaturation at 98 °C, 35 or 40 PCR cycles were carried out as follows: 10 s at 98 °C, 20 s at 60 °C, 30 or 45 s (according the couple of primers used) at 72 °C, and a final extension of 10 min at 72 °C. Amplicons were treated with the ExoSAP-IT PCR product cleanup (Applied Biosystems from ThermoFisher Scientific (Waltham, MA, USA)). A Big-Dye Terminator Cycle Sequencing Kit (Applied Biosystems) and the ABI Prism 310 Genetic Analyzer (Applied Biosystems) were used for sequencing. The 32 new Egyptian D-loop sequences were deposited in GenBank (NCBI) under the accession numbers OP132255 to OP132286.

### 2.3. Construction of the Database

From the 112 Egyptian sheep breed sequences obtained in 2014 [22] and in the present work, the identical ones were removed in order to discard possibly related animals, leading to 80 sequences (17 Barki, 12 Fallahi, 12 Ossimi, 11 Rahmani, 11 Saidi, and 17 Sohagi) including the 32 new ones. Each of these 80 sequences (access numbers in the Appendix A) was taken as seed to search for similar sequences in GenBank using the Blastn algorithm. All sequences of the well-defined breeds that differed by two or less mutations from the Egyptian ones were recorded with their corresponding accession number. The breeds presenting at least two occurrences were retained to expand the database. This was enriched by choosing thin- and fat-tailed breeds with more than 5 D-loop available sequences. We also added sequences of European and Near-East Mouflons: three, four, and one for *O. aries musimon*, *O. gmelini anatolica*, and *O. gmelini ophion*, respectively. The too incomplete sequences were removed. A total of 467 sequences including the Egyptian breed ones (Table 1 and Appendix A) were aligned using the MUSCLE program implemented in MEGA X [27]. After removing the parts protruding on both sides from the longest Egyptian sequences, the final alignment was 789 bp long.

### 2.4. Phylogenetic and Data Analyses

The assignment of sequences to the haplogroups A, B, C, D, and E [4] was made according to the phylogenetic tree reconstructed by a maximum likelihood method using PhyML v.3.0 [28] implemented in SeaView v.4 [29], the HKY85 substitution model [30], a gamma-distribution (Γ) of among-site rate variation (4 discrete categories) [31], and an estimated proportion of invariant sites. For this analysis, 720 homologous sites were retained using Gblocks [32] and the best-fit model was determined according to the BIC (Bayesian information criterion) by the fast model selection option at the W-IQ-TREE online server [33].

The ML tree topology obtained was exported in the Newick format. To estimate the affinity of the Egyptian breeds, we counted the number of terminal branches implying Egyptian individuals and those of other breeds using the program Newick Extra v. 2 written in R [34,35]. Therefore, this counting was independent of the haplogroup, which has a strong effect on other distance measures such as Nei’s or Reynold’s distances implemented in Arlequin v. 3.542 [36]. The program distinguishes the kinship of type 1, in which strictly sister sequences are considered, and of type 2, where a sequence is sister to two embedded sister sequences. The generated Excel file was then treated using Non-Metric Dimensional Scaling Plot (NMDS) and cluster analysis with user similarity and correlation as the distance metrics, respectively, implemented in PAST software v. 4.03 [37].

The *F*st values expressing the differentiation between breeds were calculated using ARLEQUIN v.3.542 [36]. After removing the breeds and species with less than three individuals (*O. gmelini ophion*, Tuj and Karadi) and the loci with more than 5% of missing data, the final file contained 34 populations over an alignment of 720 bp. The *F*st obtained values were visualized through cluster analysis with the same options as above.

## 3. Results

### 3.1. Distribution of Haplogroups in the Different Breeds

The dataset of 467 sequences was submitted to a phylogenetic analysis and the resulting tree is shown in Appendix A. From this tree, the haplogroup of each individual was deduced. As shown in Table 2, haplogroup B was dominant in the European, Egyptian, and East tropical African breeds, while haplogroup A was the most present in Eastern Asia. In Western and Central Asia, the haplogroups A, B, and C on one hand, and A and B on the other, had about the same importance, respectively.

### 3.2. Geographical Distribution of Populations and Fst Values

The *F*st values expressing the differentiation between populations were first submitted to an NMDS analysis to illustrate their distribution on the first plan (Figure 1). They identified two distinct sets of breeds overlapping inside each set. The first was on the left side of the projection, composed of Egyptian, European, and East Tropical African breeds (i.e., with a dominance of haplogroup B). Among the Egyptian breeds, the overlap with the European breed was due to Barki, Ossimi, and Rahmani. The second one on the right comprised a large envelope of Western Asian breeds, where haplogroups C, D, and E were abundant, and on the upper part, those of Eastern and Central Asia, characterized by their relatively high and low proportions in haplogroups A and C, respectively. In the corresponding cluster analysis (Appendix A), there was a set of four Egyptian breeds (Fallahi, Ossimi, Barki, and Rahmani) associated with the European and East Tropical African breeds, corresponding to the left part of the projection of the NMDS first plan, and two separated breeds (Saidi and Sohagi) close to the Cyprus fat-tailed and Kabashi on one hand, and the Lanzhou large-tailed on the other, respectively, corresponding to the right part.

### 3.3. Phylogenetic Approach of the Relationships between Breeds

To estimate the relationship between Egyptian breeds with the others using the phylogenetic approach, the calculation relative to type 1 (direct branching) from the Newick Extra program provided little information as in most cases, only one connection was revealed. In contrast, type 2 connections provided a richer dataset that was submitted to a cluster analysis (Figure 2).

As a whole, there was no structuration according to the tail phenotype or the geographic distribution. The Egyptian breeds were scattered into five parts. The Rahmani was distantly related to a group of mainly Eastern Asian distribution (Han large-tailed, Lanzhou large-tailed, Altay, Tibetan). The Ossimi was linked to two fat-tailed breeds of Western Asia (Morkaraman and Daglic). The Barki and Fallahi were associated with another group of fat-tailed breeds from Western Asia (Karadi, Karakas and Cyprus fat-tailed). The Sohagi was associated with the fat-rumped Karakul from Central Asia, while the Saidi to the four East Tropical African breeds and to the Merino from Europe.

### 3.4. Quantification of Affinities between Egyptian and Other Breeds

To substantiate the most phylogenetically linked breeds to the Egyptian ones, we report in Figure 3 the number of type-2 connections corresponding to around 75% of the total (the complete table implying 27 breeds is given in Appendix A). The connections between the Egyptian breeds themselves are treated below. The results are expressed in percentage, but the exact numbers of the connections are indicated above each histogram, revealing very contrasting situations: very few connections for Rahmani (8) but many for Saidi and Sohagi (95). For the sake of clarity, several breeds of the same region were associated: the European Waldschaf and Swiniarka, the East African Tropical Buzaei, Kabashi and Red Maasai, and the close Turkish Akkaraman and Norkaraman [38]. The fat-tailed and rumped-tailed Norduz, Edilbai, and Lanzhou (Western, Central, and Eastern Asia) and the thin-tailed Merino and Hemsin (Europe and West Asia) were considered separately.

It revealed that the two “primitive” thin-tailed breeds, Swiniarka and Waldschaf, shared a substantial proportion of links with Egyptian breeds, notably in Fallahi and Barki, in contrast to Ossimi, Saidi, and Sohagi, which showed fewer or no connection (see discussion about the term “primitive”). With regard to the links with fat-tailed breeds, their importance differed among the Egyptian breeds: Akkaraman/Morkaraman was dominant in Ossimi, but very minor in other Egyptian breeds. The Turkish Norduz and the Central Asian Edilbai harbored approximately similar percentage values in Saidi, Sohagi, and Fallahi, but there was a lack of Edilbai connection in Barki. The East Tropical African Kabashi, Buzaei, and Red Maasai had the highest number of connections with Saidi (50%), but in the range of 6 to 19% in Ossimi, Fallahi, Sohagi, and Barki. The case of Rahmani was treated separately, given its very low number of connections. However, it revealed no association with any fat-tailed breed, but two with the fat-rumped breed Hemsin.

From these results, it can be expected that the number of connections between the Egyptian breeds themselves were not uniform. The most connected breeds were the Sohagi and the Fallahi with 13 links, followed by the Saidi, which shared seven and eight links with the two previous ones, respectively. The remaining breeds only presented 0 to three associations (Appendix A). In summary, the affinities of Egyptian breeds revealed various repertoires of related breeds. There was only one couple of breeds that were strongly related: the Sohagi and Fallahi.

Following the general spread of sheep populations from Western Asia to several territories including Egypt, we can explore whether several sheep populations in this country could have been ancestors to ever southward or westward breeds. The case of Saidi revealed its relationship to East Tropical African breeds (Figure 3). For this reason, the connections of these breeds between them and with the other breeds have to be analyzed.

As explained before, there was a strong link between the fat-tailed Saidi and the thin-tailed breeds Buzaei and Kabashi (Figure 4). As for the other Egyptian breeds Sohagi and Fallahi, their part was almost as important in the Ahaamda and Red Maasai. The connections of the European thin-tailed breeds, “primitive” (Waldschaf and Swiniarka) or not (Merino), were noticeable to Ahaamda but very weak to Buzaei. The potential contribution of the Ossimi breed to the African Tropical breeds was negligible. The West Asian Hemsin (with fat limited to the base of the tail) was also connected to the Red Masai, and to a lesser extent, to Buzaei and Kabashi.

If we look at fat-tailed breeds sheltered in countries in the western direction from Egypt, there were the South Italian Laticauda, the Libyan Barbary, and the Tunisian Barbarine. As shown in Appendix A, the number of connections to the Italian Laticauda in our dataset was low due to the limited sample of this last breed. However, it revealed the noticeable influence of the European Merino, the crossed breed Assaf, the Egyptian Barki, and the East tropical African Kabashi. Regarding the Libyan Barbary and Tunisian Barbarine breeds, no D-loop sequences were currently available.

## 4. Discussion

As above-mentioned, the domesticated breeds considered here were the result of the second expansion event highlighted by Chessa et al. [5]. With regard to the origin of Egyptian sheep populations, two hypotheses were outlined in the introduction. According to the first, following the findings of archeologists [12,13], (i) the Egyptian sheep population was originally thin-tailed and (ii) the fat tail phenotype was the result of a later introgression from fat-tailed animals. According to the second hypothesis, the Egyptian breeds were supposed to originate directly from fat-tailed breeds brought by breeders, probably issued from Western Asia. From our data, can one of these hypotheses be verified for all or part of the Egyptian sheep breeds today? Moreover, did some of them contribute to the founding of breeds in other countries?

As seen before, the conclusions given by the *F*st approach were biased for the reason of the constraint of haplogroups: breeds with similar proportions in their haplogroup repertoire tend to be associated, independently of their true relationships. This is the reason why a phylogenetic derived approach was conducted, taking into consideration the number of connections between animals. This method was applied with success in the case of Moroccan and West Mediterranean breeds [34,39].

The repertoire of foreign breeds associated to each Egyptian one, depicted by our approach, was supported by other works using random amplified polymorphic DNA markers [40] and microsatellites extracted from the animals of five Egyptian breeds [25]. Both studies found that Ossimi and Rahmani significantly differed from three close breeds (Barki, Sohagi and Saidi). In this three-breeds group, Saidi was considered slightly apart in [25], which could be due to its high connection with East Tropical African breeds with a thin tail. The distinction between Ossimi and Rahmani could be explained by the strong influence of Akkaraman/Morkaraman in Ossimi, absent in Rahmani. As for the group of Barki, Sohagi, and Saidi, they shared most of the associated foreign breeds.

There were signatures supporting Hypothesis 1, as a substantial number of connections between Fallahi and Barki, and to a lesser extent, Saidi and Sohagi, to the two “primitive” European thin-tailed breeds, the Waldschaf and the Swiniarka. As reported by Gáspárdy et al. [41], the Waldschaf is “primitive” in the sense that this population is one of the descendants of the extinct Zaupel, which could be traced back to the Neolithic such as the peat sheep (*Ovis aries palustris*) [42]. The Polish Swiniarka, nearly extinct and integrated in a preservation program [43], belongs to the list of European primitive sheep breeds [44]. Curiously, we have no explanation for the connection between most Egyptian breeds and the Merino, probably due to its complex history [39,45]. Of course, the links involving both primitive European ones concern the ancestors of these breeds (i.e., a very old branch in the course of domesticated sheep). Given the similarity between the foreign repertoire recorded in the mtDNA of a group of four breeds (Saidi, Barki, Sohagi, and Fallahi), we hypothesize that these four breeds could derive from this ancestral thin-tailed population. From the present data, the most probable breeds that introduced the fat tail phenotype into the ancestral thin-tailed population would be the ancestors of the Akkaraman, Morkaraman, Hemsin, Norduz and Lanzhou large-tailed breeds. The order in which these introgressions occurred is unknown.

In contrast, the Ossimi is the most probable case supporting Hypothesis 2, given its repertoire of links to foreign breeds. We assume that the ancestors of this breed that entered into Egyptian territory were fat-tailed, issued from ancestors of West Asian Akkaraman and Morkaraman, perhaps more recently than the ancestors to the group of Saidi, Barki, Sohagi, and Fallahi.

Regarding the Barki breed, its original repertoire, marked by an absence of links to the Lanzhou large-tailed, Edilbai, Akkaraman and Morkaraman breeds, could be interpreted in two ways. The first is that a genetic drift may have occurred in Barki, in relation to its historical geographical isolation in the northwestern part of the country [46]. In contrast, this absence of links with the four cited breeds could be due to a geographical isolation occurring before the arrival of the ancestors of these four breeds. Otherwise, it has been stated that Sohagi appeared as a mixture between Barki and Saidi [25], which is quite compatible with the foreign repertoire reported in this work.

One important issue is that we faced missing or incomplete sequence data corresponding to numerous breeds. We suspect that the very low amount of connections of the Rahmani with other breeds could be due to the lack of samples of still living populations or their definitive extinction. The conclusions brought are thus limited and could be challenged with increasing material. As a result, it is difficult to test the report of Elshennawy [19], who mentioned its introduction into the territory during the 19th century from South Turkey or North Syria. As a potential candidate to explore in the future, the Awassi breed is interesting. However, most of its available sequences were too short to be included in this study, resulting in its low number of samples inside our dataset, the majority of which come from Pakistan. We hypothesize that this breed, which is largely spread in the Near East, could have some ecotypes [47] connected to Egyptian breeds, as found by Ahbara et al. [11] through their 50K SNP BeadChip analyses, and more specifically to Rahmani and Barki [48].

Regarding the influence of Egyptian breeds on other ones living in foreign countries, we found that the Barki could have contributed to the founders of the European Laticauda in the south of Italy. This result has to be taken with caution, given the low number of Laticauda sequences. Moreover, a previous work using the mitochondrial control region sequence [23] did not detect any relationship between these two breeds, but the methods were different. Nevertheless, we hypothesize here that the link between Laticauda and Barki could involve one or several intermediaries such as the fat-tailed Barbary and/or Barbarine, and such an idea deserves to be explored.

Concerning the relationships with thin-tailed breeds from East Tropical Africa, the case of the Saidi breed was unexpected. The explanation is complex and different scenarios can be drawn. The ancestors of the Saidi could have migrated southward and been the founders of Sudanese populations that were further introgressed by thin-tailed animals. The opposite view would consist of a northward migration from thin-tailed Sudanese animals with the acquisition of the fat tail occurring later, through the cross breeding with Egyptian populations. This last view is supported by the study of Abied et al. [49] using the Illumina 50K SNP BeadChip. They found that among the thin-tailed desert breeds of Sudan, the Ahaamda was clearly distinct from the others, which could correspond to the absence of the influence of Saidi in that breed. As a result, there is a set of Sudanese desert breeds (including Kabashi and Buzaei) that could have played the role of founders to the Saidi population.

Given this intricate situation, it seems comprehensible that previous authors have contradictory advice on the relationships between Egyptian sheep breeds and other breeds as well as between themselves. For example, in their study of fat-tailed breeds of Egypt and surrounding countries [50], two independent stocks could be distinguished using the Ovine 50K SNP microarray. The Egyptian animals (Barki, Farafra, Saidi, Sohagi) constituted a cluster separated from the other one formed by East African breeds (Menz and Red Maasai) settled on tropical highlands and Western Asian breeds from Cyprus (Cyprus fat-tailed), Iran (Afshari, Moghani, Qezel), and Turkey (Karakas, Norduz, Sakiz). In 2019, using the same approach but another dataset, a more gradual picture was described, tracing transitions between the Egyptian Barki and Ossimi, East African breeds (Hamrani, Kabashi), Algero-Libyan breeds (Barbarine, Berber, Libyan Barbary), and Arabian Peninsula breeds (Najdi, Omani) [11].

## 5. Conclusions

In the present work, the archeologist view of the settlement of domesticated sheep in Egypt was tested. More precisely, it was challenged whether the first animals established in the territory were thin-tailed, followed by fat-tailed ones. If this is the case, a substantial proportion of their genetic background should contain genotypes of “primitive” thin-tailed breeds. This implies that their fat tail nowadays recorded came from other breeds, which can be identified by their signature. Our approach, based on a program derived from a phylogenetic tree, support this vision for some breeds but not all. As a result, our interest was to show the events of crossings that shaped the genetic background of each breed, and therefore provided an explanation on their kinship, established by microsatellite or other methods. In the case of Ossimi, the major genetic component was linked to the present-day fat-tailed Akkaraman from Turkey. The case of the Saidi breed is also special as it was strongly linked to two Sudanese thin-tailed breeds, suggesting a common origin but through an enigmatic scenario with the present data.

It should be interesting to test our findings by extending the sampling to unexplored Libyan and Egyptian breeds, and through the investigation of ancient DNA taken from archeological records, in particular to identify the first fat-tailed breeds and their date of arrival in the territory of Egypt.

## Figures and Tables

**Figure 1 animals-12-02738-f001:**
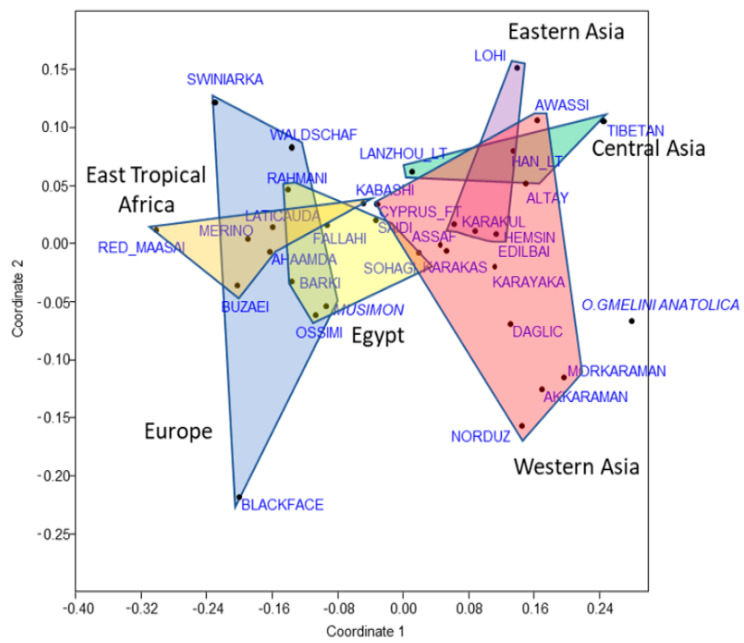
The NMDS position of the breeds based on their *F*st. The envelopes gathering the domestic *Ovis aries* breeds were drawn according to the six regions defined in Table 1. The Asian Mouflon *Ovis gmelini anatolica* was treated separately.

**Figure 2 animals-12-02738-f002:**
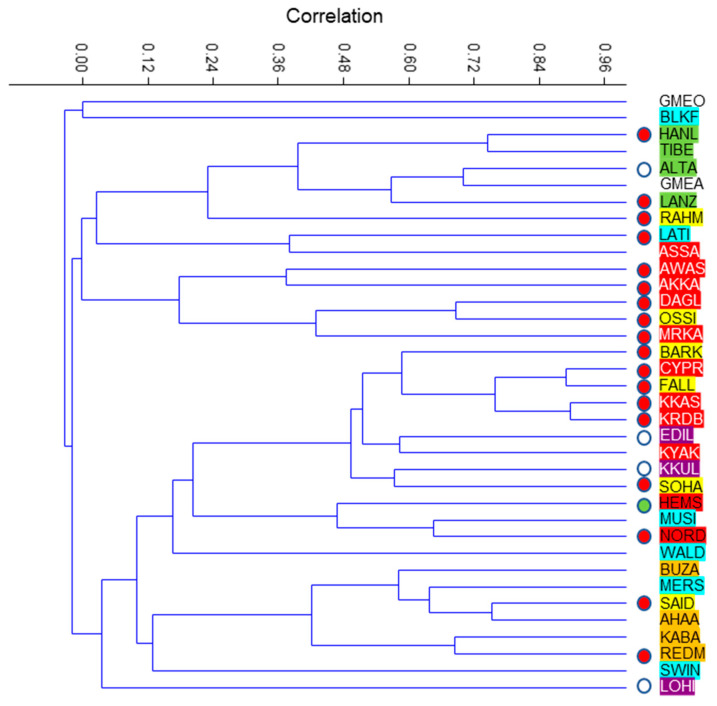
Cluster analysis from the phylogenetic analysis using Newick Extra according to the geographical distribution and tail phenotype. Wide circle: rumped tail; red circle: fat tail; green circle: fat deposit at the base of the tail; no circle: thin tail. Background color of characters: yellow: Egypt; orange: East Tropical Africa; blue: Europe; red: Western Asia; violet: Central Asia; green: Eastern Asia; white: *O. gemlini anatolica* and *O. gmelini ophion*. The breeds are noted according to the four-letter code in Table 1. The Tuj animal does not appear because it has no connection to any breed.

**Figure 3 animals-12-02738-f003:**
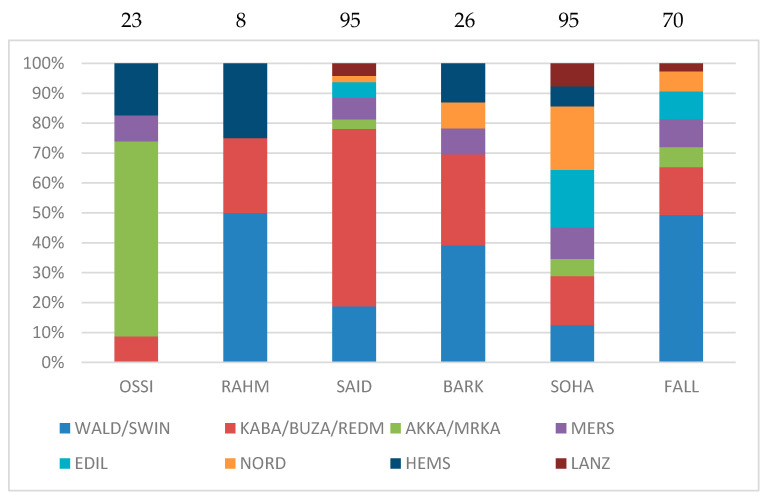
Type-2 connections between Egyptian and other breeds (77% of the total links were retained). The results are given in percentages but the sum of links for each breed is written at the top of each bar. The four-letter code of each breed is the same as in Table 1.

**Figure 4 animals-12-02738-f004:**
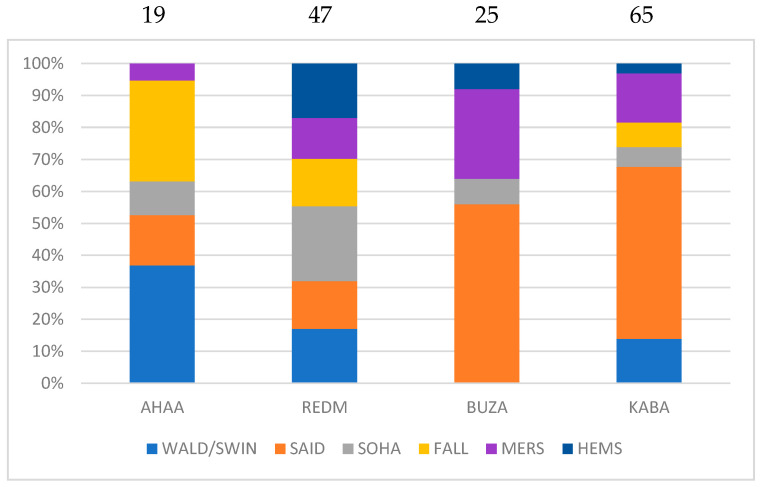
Type-2 connections between East Tropical African breeds and other breeds (around 70% of the total links were retained). The full data are presented in Appendix A. The values are given in percentages but the real sums are indicated at the top of each bar. The four-letter code of each breed is the same as in Table 1.

**Table 1 animals-12-02738-t001:** Domestic and wild *Ovis* populations of the database. The underlined characters correspond to fat-rumped breeds. The double-underlined Hemsin means a deposit of fat limited to the base of the tail. The population sizes in the present work and alternative spellings are indicated in brackets. The abbreviations used in the following are designed by a 4-letter code in upper case. The localization of the different populations was synthetized in Appendix A.

	Egypt		East Tropical Africa	Europe		Western Asia			Central Asia	Eastern Asia
**Fat-tailed breeds**	Barki BARK (17)	Fallahi (Fellahi) FALL (12)	Red Maasai REDM (20)	Laticauda LATI (5)		Akkaraman AKKA (6)	Awassi AWAS (11)	Cyprus fat tailed CYPR (8)	Edilbai (Edilbay) EDIL (11)	Altay ALTA (18)
	Ossimi OSSI (12)	Rahmani RAHM (11)				Daglic DAGL (19)	Hemsin HEMS (16)	Karadi KRDB (2)	Karakul KKUL (11)	Han large tailed HANL (18)
	Saidi SAID (11)	Sohagi SOHA (17)				Karakas KKAS (4)	Morkaraman MRKA (21)	Norduz NORD (16)		Lanzhou large tailed LANZ (18)
						Tuj TUJ0 (1)				
**Thin-tailed breeds**			Ahaamda (Al Ahamda) AHAA (16)	Latxa Black face BLKF (19)	Merino MERS (20)	Assaf ASSA (4)	Karayaka KYAK (14)	*Ovis gmelini anatolica* GMEA (4)	Lohi LOHI (5)	Tibetan TIBE (17)
			Buzaei (Buzee) BUZA (20)	*Ovis aries musimom* MUSI (3)	Swiniarka SWIN (19)	*Ovis gmelini ophion* GMEO (1)				
			Kabashi KABA (20)	Waldschaf (Waldshaf) WALD (20)						

**Table 2 animals-12-02738-t002:** Distribution of the haplogroups in the breeds in the different regions.

	Hapl A	Hapl B	Hapl C	Hapl D	Hapl E
East Tropical Africa	7	69	0	0	0
Egypt	11	63	5	0	1
Europe	4	82	0	0	0
Western Asia	37	39	38	4	8
Central Asia	10	13	4	0	0
Eastern Asia	38	29	5	0	0

## Data Availability

The new D-loop sequences were deposited in GenBank (NCBI) under the accession numbers OP132255 to OP132286. These are integrated in Appendix A containing all of the accession numbers of the 467 sequences used in this work.

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
