# Peer review of "Shedding Light on the Origin of Egyptian Sheep Breeds by Evolutionary Comparison of Mitochondrial D-Loop"

_animals, 2022, doi:10.3390/ani12202738_

Round 1

Reviewer 1 Report

The authors presented an interesting paper on the origin of Egyptian sheep breeds by evolutionary comparison of mitochondrial D-loop. In my opinion the investigation is well conducted and the paper is well written. I have only some minor comments aiming to further improve the manuscript.

Keywords: it could be better to avoid words already included in the title.

Line 63: it could be better “As demonstrated through an investigation carried out using SNP markers….”

Material and methods: it could be very useful for the readers to add (in the manuscritp or as supplementary materials) an image including pictures of the studied breeds that are not known to all the readers. One animal is enough (ram and ewe it is also better).

Line 111: I am confused about the number of samples used for the study (and the total number of sequences (467 in line 174). Can you please add a sentence to better explain how many samples came from previous studies and how many samples were collected for this study? And how many sequences from Genebank were added to the sample? Were the samples from males and females? Please specify.

Line 113: the used primers were tested in a previous investigation? If so it is mandatory to include the reference. If setted up for this study, specify the software used for the primers design.

Line 120: please replace DNTP with DNTPs.

Line 132: please specify the reference sequence used in the blast procedure.

Line 170: please write Fst putting “st” as subscript and “F” in italic style (in all the manuscript long).

Line 182: double “p” in populations

Figure 1 title: please use italic style for latin names.

Author Response

Keywords: it could be better to avoid words already included in the title.

We changed some of them and proposed the new set of keywords:

Fat-tail introgression; Mitochondrial control region; Egyptian breed; Phylogenetic analysis.

Line 63: it could be better “As demonstrated through an investigation carried out using SNP markers….”

Done

Material and methods: it could be very useful for the readers to add (in the manuscript or as supplementary materials) an image including pictures of the studied breeds that are not known to all the readers. One animal is enough (ram and ewe it is also better).

We referred to two papers that include pictures of 5 Egyptian breeds among the 6 studied here (Galal et al., 2005; De Pauw et al., 2011). The remaining Fallahi was illustrated by pictures of ram and ewe taken by Pr Othman in a new supplementary figure S1. A Barki ram picture was added.

Line 111: I am confused about the number of samples used for the study (and the total number of sequences (467 in line 174). Can you please add a sentence to better explain how many samples came from previous studies and how many samples were collected for this study? And how many sequences from Genebank were added to the sample? Were the samples from males and females? Please specify.

We used the genomic DNA extracts sampled by Othman et al. in 2014 [22] and 2018 [24] from blood of animals of both sexes to achieve new sequencing.

From the 112 Egyptian sheep breed sequences obtained in 2014 [21] and in the present work, the identical ones were removed in order to discard possibly related animals, leading to 80 sequences (17 Barki, 12 Fallahi, 12 Ossimi, 11 Rahmani, 11 Saidi, and 17 Sohagi), including the 32 new ones….

A total of 467 sequences including Egyptian breed ones (supplementary table S1) were aligned using MUSCLE program implemented in MEGA X [27]. After removing the parts protruding on both sides from the longest Egyptian sequences, the final alignment was 789 bp long.

Line 113: the used primers were tested in a previous investigation? If so it is mandatory to include the reference. If setted up for this study, specify the software used for the primers design.

We specified the origin of the used primers.

For this purpose, PCR were performed using the combination of the forward primers, CR0F (5’-TGGTCTTGTAAACCAGAGAAGGAG-3’) designed using Primer-Blast from NCBI tools, or CR2F (5’-GAAGTTCTACTTAAACTATTCCCTG-3’) and the reverse one CR3R (5’-GATGCTCAAGATGCAGTTAAGTCC-3’), both primarily designed in [26].

With [26]: Tapio, M.; Marzanov, N.; Ozerov, M.; Ćinkulov, M.; Gonzarenko, G.; Kiselyova, T.; Murawski, M.; Viinalass, H.; Kantanen, J. Sheep mitochondrial DNA variation in European, Caucasian, and Central Asian areas. Mol. Biol. Evol. 2006, 23(9), 1776-1783.

Line 120: please replace DNTP with DNTPs.

Done

Line 132: please specify the reference sequence used in the blast procedure.

Each one of these 80 sequences (access numbers in table S1) was taken as seed to the search of similar sequences in GenBank using Blastn algorithm.

Line 170: please write Fst putting “st” as subscript and “F” in italic style (in all the manuscript long).

Done

Line 182: double “p” in populations

Done

Figure 1 title: please use italic style for latin names.

Done

Reviewer 2 Report

In this manuscript, in order to describe the genetic relationships between several domestic sheep populations in North African Maghreb countries, East Tropical Africa, Arabian Peninsula, and Near East countries. The authors used a phylogenetic-derived program to compare the control region of mtDNA of many breeds with fat or thin tails, but there are many limitations in this study. The comments are as follows:

(1) In the simple summary, line 17 showed 34 breeds were used in this study, but in the abstract (line 26), 37 breeds were described.

(2) The sample in each breed was too small, some breeds only a few.

(3) The author's style of writing does not conform to paper writing, especially in the Materials and Methods, I suggest the authors draw up subheadings and write them in sections.

(4) Table 1, please use a three-line table.

Author Response

In this manuscript, in order to describe the genetic relationships between several domestic sheep populations in North African Maghreb countries, East Tropical Africa, Arabian Peninsula, and Near East countries. The authors used a phylogenetic-derived program to compare the control region of mtDNA of many breeds with fat or thin tails, but there are many limitations in this study. The comments are as follows:

  • In the simple summary, line 17 showed 34 breeds were used in this study, but in the abstract (line 26), 37 breeds were described.

The number of 34 breeds has been corrected to 37 breeds in the simple summary. However, 34 breeds were analyzed during Fst calculations, whereas there were 37 during phylogeny and derived analyses.

  • The sample in each breed was too small, some breeds only a few.

It is quite true. About the Egyptian breeds, we have explored the totality of available sequences in NCBI GenBank sharing a satisfactory length, in particular in their 5’ end relatively to our sequences. For most foreign domesticated breeds, we retained 18-21 animals as a limit number, not to have an over-representation relatively to Egyptian animals. As regard the remaining breeds, we included from GenBank the maximum number of animals fulfilling our criteria. The Tuj breed was interesting because it is a fat-tailed breed from Western Asia. Otherwise, among the different sequences, only one had a satisfactory length. The non-domesticated animals (O. aries musimon, O. gmelini anatolica and O. gmelini ophion) were considered because they are markers of Ovis evolution, but not to retrieve a connection with Egyptian breeds, justifying a very small population size.

  • The author's style of writing does not conform to paper writing, especially in the Materials and Methods, I suggest the authors draw up subheadings and write them in sections.

We added in the Materials and Methods section the subheadings: Animals – Sequencing – Construction of the Database - Phylogenetic and Data Analyses.

  • Table 1, please use a three-line table.

Done

Round 2

Reviewer 2 Report

1.  table 1 and table 2, please use a three-line table. 

2.  the English writing should be improved.

3. I still think there are many limitations in this study, and it is not suitable for published in Animals.